# Secretory Autophagy Forges a Therapy Resistant Microenvironment in Melanoma

**DOI:** 10.3390/cancers14010234

**Published:** 2022-01-04

**Authors:** Silvina Odete Bustos, Nathalia Leal Santos, Roger Chammas, Luciana Nogueira de Sousa Andrade

**Affiliations:** Center for Translational Research in Oncology (LIM24), Instituto do Câncer do Estado de São Paulo, Hospital das Clínicas da Faculdade de Medicina da Universidade de São Paulo, Sao Paulo 01246-000, Brazil; nathalialeal@usp.br (N.L.S.); rchammas@usp.br (R.C.)

**Keywords:** secretory autophagy, exosomes, secretion, melanoma, tumor microenvironment, tumor resistance

## Abstract

**Simple Summary:**

Tumor microenvironment (TME) is a complex of many cell types and extracellular matrix that play an active role in regulating and sustaining melanoma tumor progression. In this context, the secretion of several molecules, by secretory autophagy or exosome release, stimulates the intercellular communication between the different components of the TME modulating tumor response. Here, we discuss the current awareness around the role of extracellular secretion in melanoma TME and also investigate the molecules related to these secretion pathways in melanoma progression using public databases.

**Abstract:**

Melanoma is the most aggressive skin cancer characterized by high mutational burden and large heterogeneity. Cancer cells are surrounded by a complex environment, critical to tumor establishment and progression. Thus, tumor-associated stromal components can sustain tumor demands or impair cancer cell progression. One way to manage such processes is through the regulation of autophagy, both in stromal and tumor cells. Autophagy is a catabolic mechanism that provides nutrients and energy, and it eliminates damaged organelles by degradation and recycling of cellular elements. Besides this primary function, autophagy plays multiple roles in the tumor microenvironment capable of affecting cell fate. Evidence demonstrates the existence of novel branches in the autophagy system related to cytoplasmic constituent’s secretion. Hence, autophagy-dependent secretion assembles a tangled network of signaling that potentially contributes to metabolism reprogramming, immune regulation, and tumor progression. Here, we summarize the current awareness regarding secretory autophagy and the intersection with exosome biogenesis and release in melanoma and their role in tumor resistance. In addition, we present and discuss data from public databases concerning autophagy and exosome-related genes as important mediators of melanoma behavior. Finally, we will present the main challenges in the field and strategies to translate most of the pre-clinical findings to clinical practice.

## 1. Introduction

Melanoma is a type of skin cancer developing from the melanocytes. Despite being a rare skin cancer, it is the most aggressive due its heterogeneity and high mutational burden [1]. Although several therapies are available for patients, the treatment to advanced melanoma patients is still poor and is frequently associated with side effects and acquired resistance [2,3]. In this regard, autophagy machinery has been flagged as a resistance mechanism of melanoma cells to alleviate metabolic stress induced by chemotherapeutic drugs; also, it has been considered an important component in melanogenesis and melanoma progression [4,5,6,7].

Autophagy is a self-digestion process that contributes to nutrient recycling and quality control by lysosome-dependent degradation of damaged organelles and proteins. In physiological conditions, autophagy is necessary to maintain the cell homeostasis; however, in stress conditions, such as those in which tumor cells are exposed, autophagy is usually induced as a self-protective mechanism. Nevertheless, autophagy has a dynamic role in cancer, being involved both in tumor prevention and promotion [8]. Furthermore, it is established that autophagy has a crucial role within the tumor microenvironment (TME). The TME is a complex piece of cancer, consisting of stroma cells and components of the extracellular matrix, critical to determinate tumor fate. The physiological features of the TME, such as nutrient deprivation, acidity, hypoxia, and inflammation are stressors capable of induce autophagy, which may modulate tumor progression and therapy outcome. Thus, exploring the autophagy system in melanoma TME is extremely important for discovering new therapies and enlightening the mechanisms that regulate tumor malignity. Therefore, besides the classical degradative function, novel autophagy roles were described in the last decade. Recent findings related to autophagy mediating unconventional secretion demonstrate that cancer cells can communicate with the surrounding cell by an autophagy-dependent secretion named secretory autophagy (SA), which changes functions of immune cells, induces drug resistance, and drives an invasive behavior [9,10,11]. Interestingly, this new secretory function of autophagy reveals an overlap between autophagy and the vesicular membrane trafficking. In this review we will focus on secretory autophagy and the interplay with exosome biogenesis within the melanoma TME.

## 2. Secretory Mechanisms: A Complex Network for Cargo Release

Living cells make use of secretion pathways to deliver intracellular components to the extracellular milieu. Secreted factors released outside can be part of a clearance mechanism or also mediators of intercellular communication, both in normal and cancer cells, playing physiological and pathological roles, respectively. One example of such a mechanism in normal cells is the transfer of melanosome vesicles from melanocytes, cells from which melanoma derives [12]. Considering this primary function of melanocytes and knowing the stromal mimicry by melanoma cells, here we emphasize and explore the influence of the secretory mechanisms used by melanoma cells and their impact within the TME [13].

### 2.1. Secretory Autophagy

This process is a non-degradative function of autophagy, where the cells control the release of material to the extracellular environment. Although this mechanism was observed years ago, it is only now well recognized as a novel secretory pathway. In the last few years, many studies uncovered the molecular machinery involved in secretory autophagy; however, several issues have not been explained yet, in part due to the tight proximity with the classical autophagy and the endosomal machinery. However, some authors defend that as SA it is not a degradation mechanism, it should not be termed as an autophagy (“self eating”) process, but the participation of several classical autophagy components in SA route raise questions about whether it can be considered an entire different pathway or just an alternative route. The same difficulty emerges from SA and exosomes, which are a subtype of extracellular vesicles (EVs) secreted by cells that will be better discussed below. Since they have common proteins involved in their biogenesis, doubts about whether they represent different branches of a single mechanism rather than two independent pathways are still not solved [14,15].

Secretory autophagy (SA) is an independent ER-Golgi route that supports the traffic of a growing list of proteins that use unconventional secretory mechanisms and proteins that are redirecting to this pathway, from classical to unconventional secretion, under stress conditions [16]. One of the first molecules secreted by the secretory autophagy pathway was the pro-inflammatory cytokine IL-1β [17]. Because SA secretes molecules like IL-1β, IL-18 and HMGB1, initially it was related to inflammatory processes; however, later studies described other substrates, such as Annexin-I and Galectin-3 [18].

Briefly, in this process the cargo is not ubiquitinated to be degraded in the lysosomes; instead, the proteins in the autophagosomes are secreted out after their fusion with the multivesicular bodies (MVBs) to form amphisomes, which are then fused to secretory lysosomes or direct to the plasma membrane [19,20]. Notice that degradative and secretory autophagy share several molecules and events, such as the formation of the intermediate organelle and the amphisome, before the fusion with the lysosomes or the plasma membrane, respectively.

SA requires several components of the autophagy system, such as ATGs and SNAREs, which are involved in the fusion membrane, and the Rab-GTPase family proteins related to vesicular trafficking. The first studies examining the source of the membranes involved in the assemblage of secretory autophagosomes were conducted in yeast upon starvation; the authors indicated the association of membrane microdomains rich in PI3P, called the compartment for unconventional protein secretion (CUPS), with the initiation of autophagosome formation [21]. In mammals, the initial phase of the SA remains unclear, but it is supposed that the biogenesis of secretory autophagosomes occurs from an omegasome-like structure, equivalent to CUSP (Figure 1). In this step, the participation of GRASP65 and GRAP55 (Golgi Reassembly Stacking Proteins) was found [22,23]. In addition, a few authors demonstrated that multiple autophagy proteins, such as ATG7, ATG3, ATG5, ATG12, ATG16LI, and LC3II, play a significant role in autophagy secretion [24]. One example is the study of Young and Cols that noted a link between autophagy and senescence describing autophagy activation as an effector mechanism of the senescence responsible for the senescence-associated secretion of IL-6 and IL-8. As the mRNA levels of both interleukins were higher in *Atg5*/7 knockdown cells, the secretion occurs by autophagy-dependent posttranslational mechanism [25]. As well, autophagy-competent cancer cells treated with chemotherapy induced an immunogenic ATP release, unlike that observed in *Atg5*/7 knockdown cells [11]. In another work, the interaction between the ESCRT-associated protein Alix and the Atg12-Atg3 complex were necessary to maintain MVBs morphology and vesicular traffic as well as the exosome biogenesis [26].

How secretory autophagy guides the autophagosome to the plasma membrane is uncertain but latest studies reported some key elements. Investigating protein secretion upon the lysosomal damage model, Kimura et al. found specialized cargo receptors of the TRIM family. Among them, TRIM16 proved to be necessary for the secretion of IL-1β in cooperation with an R-SNARE, called Sec22B. Then, the autophagosome is addressed to the plasma membrane where it finds Qa-SNARES, syntaxin 3 and 4, and Qbc-SNAREs to form a SNARE complex responsible for the membrane fusion (Figure 1) [27]. Nevertheless, the autophagosome can follow a different path to fuse with the MVBs and give origin to the amphisomes. The latter can be degraded by fusion with lysosomes or released to the extracellular milieu [28]. Due to a hybrid origin, the amphisomes carry classical autophagy markers, such as ATG5, ATG16L1, and LC3 and endosomal markers such as Rab7, Rab11, Rab27, and Rab35 [14]. This feature indicates a connection between both pathways.

### 2.2. Branches of the Secretory Autophagy System

Recent evidence showed a novel mechanism, where LC3 is implicated in the loading of RNA-binding proteins (RBPs) and small non-coding RNAs within EVs. This process, named LC3-dependent EV loading and secretion (LDELS), requires the LC3 conjugation machinery; however, no other ATGs are involved in degradative autophagy, representing an alternate route for the EVs cargo secretion. In LDELS, LC3 and vesicular cargo are delivered to the limiting membrane of MVBs, where the cytosolic cargo undergo budding, generating intraluminal vesicles (ILVs) within the MVBs that subsequently are released as extracellular vesicles. Importantly, when these EVs were isolated and fractionated, it was observed that LC3 was in the lumen of EVs, thus corroborating that LC3 is secreted within EVs [29]. Afterwards, another elegant study from the same group revealed a different secretory autophagy route in response to the endolysosome inhibition. They showed that the inhibition of lysosome acidification by bafilomycin A1 activates secretory autophagy of autophagy elements and cargo receptors in EVs fractions, both in vitro and in vivo [30]. In contrast to LDELS, the cargo is not loaded selectively into EVs, being exposed to proteolysis instead. Thus, these proteins are released in association with EVs, but as a fraction of nanoparticles, named extracellular vesicles and particles (EVP) [31]. Thus, this secretory pathway is dependent on several ATG as well as endosomal markers, such as Rab27a, which are responsible for the exocytosis of EVPs. In addition, the impairment of the degradative autophagy induced EVP-associated secretion maintaining cell protein homeostasis. This suggests a regulation between secretory and degradative autophagy. Otherwise, this regulation is also demonstrated in other studies; Kraya and Cols described that high autophagy levels in melanoma cells impact the level of secreted proteins, increasing cellular secretion [9]. Interestingly, these authors also observed higher levels of autophagy and protein secretion, such as CXCL8 and IL-1β, in metastatic melanomas in comparison to cells derived from primary lesions. CXCL8 has been described as an important regulator of growth, angiogenesis, migration, and metastasis in melanomas through its binding to CXCR1 and CXCR2 receptors [32,33,34]. Consistently, Scheibenbogen and co-authors (1995) demonstrated that higher levels of CXCL8 secreted by melanoma cells correlate with poor prognosis. Concerning IL1-β, using a pre-clinical model, Tengesdal et al. (2021) demonstrated that IL-1β induces IL-6/STAT3 activation in melanoma, inducing an immunosuppressive environment that favors tumor growth. Then, based on these findings, one might speculate that along melanoma progression, tumor sub clones demonstrating increased levels of SA and EVPs are selected due to their ability in communicating with stromal cells to imprint a permissive microenvironment for tumor growth and metastasis, highlighting the importance of unconventional secretory pathways as an important hallmark in melanoma evolution [35,36].

### 2.3. Crosstalk between Extracellular Vesicles and the Autophagy Pathway

EVs are a heterogeneous population of spheric lipid bilayer vesicles, which are secreted by most cells to biological fluids playing key roles in physiological and pathological processes. EVs contain several molecules, such as proteins, lipids, non-coding RNAs, mRNA, DNA, and cytokines protected by a double membrane. Thus, the main attribute of EVs is related to its function as mediators of intercellular communication. There are different classes of EVs, classified based on size, biogenesis, content, and function. The two groups best characterized and most studied of EVs are: microvesicles (100–1000 nm) and exosomes (30–150 nm) [37]. Owing to their endosomal origin, the exosomes share some molecules and events with SA. During the exosome generation, the cargo is sorted to invaginations of the endosomal membrane establishing ILVs within the MBVs, which fuse with the plasma membrane to release the intraluminal vesicles to the extracellular compartment as exosomes (Figure 1) [38]. The exosomes convey different kinds of molecular messengers between cells contributing with the regulation of cellular phenotypes. Then, in cancer, the interest of the tumor-derived extracellular vesicles (TEVs) has grown due to its crucial signaling role between tumor cells and the TME supporting tumor progression, tumor resistance, and their potential as diagnostic and prognostic biomarkers.

Considering the above data and the importance of the intercellular communication into the TME, recent studies try to shed light on the importance of vesicular secretion programs in melanocytes and melanoma cells. Next, we further discuss the conjunction of these pathways in melanocytes biology and in the context of melanoma tumor progression, drug response, and recurrence.

## 3. Role of Secretory Autophagy and Exosomes in Melanocytes Biology

Although they are still described as different cellular secretory routes, the similarities between autophagy and exosomes have been reported by some groups and, in fact, are also reflected in their history. In the past years, for example, both processes were seen as a way for cells to get rid of waste or undesirable molecules [39]. However, over the years, the understanding about their biology revealed that both cellular processes are also involved in homeostasis maintenance and cell-to-cell communication. Regarding the common molecules, HSPA8, HSP90AA1, VCP, Rab7A, Rab8a, GRASPs, LC3, ESCRTs, SNAREs, and several Atg proteins are the main ones described [17,40,41,42].

Autophagy and exosomes machinery have been reported to play an important role in skin melanocytes. These cells are the ones responsible for pigment production through the conversion of tyrosine into melanin, in a multi-step process that relies on intense intracellular vesicle trafficking, which culminates in melanosome formation. Exosome biogenesis-related molecules as well as proteins of the autophagic machinery are involved in this vesicular movement inside melanocytes in a very coordinated way, which guarantees the production and packing of melanin in melanosomes and their transfer to keratinocytes [43,44,45,46]. Furthermore, it is known that melanocytes are often exposed to stressful environmental conditions, such as UV irradiation. In this scenario, the induction of autophagy and extracellular vesicle release is reported in human cutaneous melanocytes. Shen and co-authors (2020) observed an increase in exosome release by human melanocytes after UVB exposure [47]. Regarding their cargo, an enrichment in miR-320d, miR-4488, and miR-7740 was detected by the authors, suggesting that gene expression might be affected in recipient cells. Interestingly, in the same year, Sha and Cols also demonstrated an increase in exosome secretion by melanocytes in response to UV [48]. Juvenil human melanocytes entered a premature senescent-like state, which was accompanied by a decrease in most NER (nucleotide excision repair) genes after UV exposure. The exosomes released under this condition exhibited a distinct microRNA cargo known to target genes related to senescent phenotype, and the authors suggested that these nanostructures are responsible for the senescent phenotype in irradiated melanocytes, which might favor the malignant transformation of melanocytes for leaving these cells vulnerable to irreversible DNA lesions.

In the same way, autophagy has been reported as an important mechanism for skin homeostasis and, consequently, is involved in processes related to skin aging, senescence, and UV response. Regarding the latest processes, in 2016, Wäster and colleagues demonstrated that UVA induced plasma membrane damage in melanocytes and lysosomal exocytosis is involved in its repair [49]. In relation to senescence, using *Atg7*-deficient skin melanocytes, Zhang and colleagues observed the induction of premature senescence in these cells, characterized by the accumulation of p62, up-regulation of NFR2 (nuclear factor E2–related factor 2), and increase in oxidative stress [50]. The induction of both autophagy and senescence phenotype in stress conditions, such as DNA damage and oxidative stress, suggest a link between them. Supporting this, Ni and Cols demonstrated an increase in inflammatory chemokines and cytokines secretion by *Atg7*-deficient melanocytes, which was associated with the senescence-associated secretory phenotype (SASP) [51]. In addition, melanocytes transduced with BRAF oncogene, representing an early-stage pathogenesis melanoma model, demonstrated that the oncogene-induced proliferation can be regulated by ATG5 expression; hence, the down-regulation of ATG5 lead to low autophagy levels allowing proliferation and preventing senescence [52]. Collectively, these studies highlight the involvement of the vesicular secretory pathways as mediators of melanocyte stress response; however, it is still not clear whether these processes act in synergism or even if they indeed constitute the same cellular pathway evoked in response to a stress condition. From our point of view, in any of these scenarios, the activation of autophagy and/or exosome release represents an effective way to survive stressful situations. We believe that this strategy is also employed by malignant melanoma cells to bypass the cytotoxicity imposed by oncological therapies, as discussed below.

## 4. Melanoma Progression, Tumor Microenvironment, and Autophagy

The environment around the tumor is a complex network of stromal cells, including endothelial cells, fibroblasts, and immune cells as well as the non-cellular components, such as the extracellular matrix and secreted factors, which are all important to stimulate tumor heterogeneity. Each element of the TME has essential roles to control several stages of tumorigenesis, such as tumor growth, invasion, metastasis, and neovascularization. Among them, cancer cells and tumor-associated stromal cells are crucial in altering the equilibrium of extracellular matrix remodeling events, providing a dynamic niche capable of sustaining malignant transformation [53]. In turn, the extracellular matrix impacts proliferation, mobility, and tumor vascularization. In the latest, changes in tissue tumor architecture, caused by the overgrowth of cancer cells, aid the establishment of the tumor hypoxia state, which ultimately induces a switch to the angiogenic phenotype. Such events lead to the sprouting of new blood vessels vital to deliver sufficient oxygen and nutrients to tumor survival [54,55]. Above all, the autophagy secretion process has emerged as a regulator of this phenomenon. For instance, the secretion of different cytokines, such as VEGF-A and IL-8 by melanoma cells, trigger pro-angiogenic signaling in surrounding endothelial cells and prompting an angiogenesis switch, which is correlated with the transition of melanocytic lesions to vertical growth phase—a stage with a higher capacity to metastasize [17,56,57].

The interplay between TME components and secretory autophagy are also crucially involved in local invasion and metastasis, important features in high resistant and metastatic tumors, such as malignant melanoma. In this context, it was demonstrated that MMP-9 and MMP-2 in melanoma patients are key players in extracellular matrix degradation and drivers of cancer cell spreading and metastasis [58,59]. These peptidases are regulated by different molecules, such as the angiogenic factor IL-8 in an autophagy-dependent secretion manner [33], reinforcing the notion that SA establishes an effective communication in the TME. Moreover, another way that autophagy contributes to TME reprogramming is through the regulation of immune cells. Based on the immunogenic phenotype of melanomas characterized by the infiltration of different immune cells, the activity of the autophagy pathway in this scenario is meaningful. Inhibition of autophagy in melanoma cells, for example, promotes NK (natural killer) cells infiltration and activation against the malignant cells. Interestingly, targeting the autophagy process in MDSC (myeloid-suppressor derived cells) reprograms these cells towards an anti-tumoral phenotype. On the other hand, the secretion of diverse factors into the TME through SA enables the induction of immune evasion and immunosuppression, affecting therapy responses and contributing resistance (reviewed in [60,61]), as better discussed in the next section. Then, taken together, these studies highlight the oncogenic role of autophagy in the TME in the context of tumor cells as well as stromal cells, and, in fact, we imagine that its blockade specifically in immune cells might improve immunotherapy response and provide clinical benefits to melanoma patients.

## 5. Autophagy and Exosome Signaling Modulates Melanoma Behavior and Therapy Outcome

Tumor recurrence after treatment is an important issue that limits treatment efficacy and often represents the cause of a patient’s death. It is known that tumor clones harboring intrinsic resistance are responsible for tumor relapse [62]. On the other hand, acquired resistance under cytotoxic treatment also represents another way that culminates in tumor outgrowth and treatment failure [63]. Several studies have been conducted to elucidate the molecular and cellular mechanisms behind this phenomenon, and, not surprisingly, exosomes and autophagy have been shown to play a significant role as inducers of resistant phenotypes [64,65,66,67,68,69]. In cutaneous melanoma, its enhanced secretory ability is in part responsible to increase its fitness to the detrimental conditions (such as acidosis, nutrient deprivation, and hypoxia) in the TME and to escape from immune system surveillance [70]. Apart from this, Alonso-Curbelo and co-authors (2014) demonstrated that, although being highly heterogeneous and plastic, melanoma cells retain developmental memory characterized by the expression of lysosome-related genes, including Rab7, among others, which is known to be also involved in exosome secretion [71,72]. The study conducted by Alonso-Curbelo and colleagues revealed that Rab7 induction occurs at early stages of melanoma development and positively regulates cell proliferation. Then, based on these findings, one can assume that vesicular genes involved in autophagy and exosome release are additionally important for melanoma survival under the cytotoxic stress imposed by therapy.

In fact, concerning the autophagic process, some groups already demonstrated that certain cytotoxic therapies induce autophagy in several tumor types. However, this process can trigger tumor cell death or promote cell survival instead, which seems to be dependent on the cell type and stressor agent. In melanomas, Ma and co-authors [73] verified that a high autophagic flux is associated with poor survival in the metastatic disease. Moreover, when analyzing tumor specimens obtained from melanoma patients treated with temozolomide and sorafenib, the same group found that patients harboring tumors with a high autophagic index were less likely to respond to chemotherapy and indeed presented shorter survival, suggesting that targeting autophagy could reverse the chemoresistance in these patients [5]. Focusing on targeted therapy, previous studies revealed that BRAF inhibition upregulates autophagy, whereas inhibition of this process overcomes BRAF inhibitor-induced resistance ([5,74,75]). However, a recent study demonstrated that fluctuations in autophagy flux occur in response to drug treatment as reported by Verykiou et al. (2019). [4] The authors detected that the resistance to the MEK inhibitor in melanoma was accompanied by an increase in autophagy in a short-term period after trametinib treatment. However, in the long term, this increase was followed by a reduction of autophagy to its basal levels in the surviving cells, indicating that harnessing autophagy can represent a way to bypass induced resistance in metastatic melanoma.

In 2019, Li and colleagues demonstrated that BRAFV600E-TFEB-ZKSCAN3-autophagy and lysosome biogenesis axis are involved in tumor resistance to targeted therapy in BRAF mutant melanomas, leading to the aberrant expression of TGF-beta pathway and impairing the tumor response to BRAF inhibition [76]. This study explored the role of both vesicular compartments in melanoma resistance, and, although the secretory pathway was not evaluated, we reasoned that it may affect the sensitivity of naive tumor cells (bystander effect). However, evidence about secretory autophagy and drug resistance in this type of tumor was provided by Martin and co-authors (2017) [77]. After the acquisition of resistance to vemurafenib, a drug that targets mutated BRAF, resistant melanoma cells exhibited increased autophagy and increased secretion of ATP. The authors observed a bystander effect evoked by ATP and its purinergic receptors in tumor cells; this effect was characterized by the maintenance of the resistant phenotype and cell migration, boosting not only tumor progression but also impairing tumor eradication. In fact, most of the studies in autophagy and BRAF-mutated melanomas focus on the role of the degradative sub pathway in the acquired resistance, which is restricted to the tumor cell. Concerning SA, as described above, there are very few studies about the role of this secretory pathway in BRAF melanomas and we believe that this route may induce resistance not only in tumor cells, but it can also reprogram other stromal cells present in the TME through the secretion of important pro-tumoral molecules. In other words, stromal cells can adopt a behavior triggered by tumor-secreted molecules upon targeted therapy, contributing to treatment failure and tumor outgrowth. Besides that, it is also reasonable to consider that important factors released by SA might reach the blood vessels and invoke cellular alterations at longer distances, producing systemic modifications that would help in the dissemination of the disease, for example.

The involvement of the autophagic process in modulating resistance is not restricted to chemo and drug-targeted therapies. Recently, using the B16F10 murine melanoma model, Kim and co-authors observed increased levels of LC3B and Nanog, a pluripotent stem cell gene, in these melanoma cells that had been previously shown to be resistant to the immune checkpoint blockade (ICB) therapy [68]. Upon the knockdown of LC3B and Nanog, B16F10 cells became more sensitive to cytotoxic T lymphocyte (CTL)-mediated apoptosis in vitro and showed a significant reduction in its sphere-forming capability. More interestingly, they showed that inhibition of LC3B was able to boost the therapeutic effect of anti-PDCD1 in vivo, overcoming the resistance to ICB therapy promoted by autophagy.

Wen et al. (2018) elegantly demonstrated that autophagosomes released by B16F10 cells promote an immunosuppressive environment through the induction of M2 phenotype in macrophages [78]. At the molecular level, these vesicles are taken up by macrophages followed by the activation of p38 and Stat3, leading to IL-10 release and upregulation of PD-L1. Consequently, the authors observed a decrease in T cell proliferation characterizing the immunosuppressive activity of melanoma autophagosomes. The involvement of secretory autophagy in the establishment of a tumor-suppressive microenvironment in melanoma was also described in the study conducted by Tzeng and co-authors (2021) [79]. Upon MitoX (mitoxantrone X) treatment, the melanoma cells secreted PAI-1 (plasminogen activator inhibitor-1) through autophagy induction. The lack of effectivity was noticed when tumor-bearing animals were treated with MitoX; however, the silencing of Beclin-1 in B16F10 cells overcomes drug tumor resistance, which was characterized by an increase in infiltrating CD8 T cells and a decrease in Foxp3 and Arginase 1 positive cells in the TME, highlighting the role of secretory autophagy in treatment response.

Nevertheless, it is also true that exosome release is involved in resistance as reported by several groups along the years in different tumor types [80]. In response to vemurafenib, exosomes released by melanoma cells exhibited a differential micro-RNAs and protein cargo which were related to drug response [81,82]. Vella et al. found that PDGFRβ (platelet-derived growth factor receptor β) is transferred from resistant melanoma cells to recipient cells through extracellular vesicles, including exosomes, and it promotes resistance via activation of PI3K/AKT signaling pathway in the latter ones [83].

The transfer of resistance by exosomes is also reported in non-tumoral cells that compose the TME, which compromise therapy efficiency. A recent study by Liu and co-authors (2021) revealed how exosomes secreted by melanoma cells can impair anti-PD-L1 therapy under this context [84]. The authors observed an increase in G2/M cell cycle arrest followed by apoptosis in response to inhibition of glutamate release by the cystine/ glutamate transporter cystine-glutamate exchange (xCT). However, when tumor-bearing mice were treated with SAS, a xCT inhibitor, altogether with anti-PD-L1/PD-1, a reduction in the effectiveness of ICB therapy was reported. Investigating the cellular and molecular mechanisms behind this effect, the authors found that exosomes secreted by melanoma cells under this combined treatment were enriched in PD-L1, inducing tumor outgrowth even in the presence of SAS. Observing the TME, the decrease in CD4/CD8 infiltrating T cells was accompanied by an increase in M2 macrophages, which are known to exert pro-tumoral activities. In particular, the anti-inflammatory M2 phenotype was induced by the exosomes carrying PD-L1, indicating that the secretome characterized by exosomes under treatment participates in tumor repopulation. In fact, these results are in accordance with our findings [85]. We also noted that vesicles, composed by exosomes and microvesicles and released by human melanoma cells in response to the chemotherapeutic drug temozolomide, promoted the M2 phenotype in macrophages and tumor outgrowth in nude mice, highlighting that stromal cell are also affected by these vesicles. Based on the effect of exosomal PD-L1 in melanoma resistance, Wang and colleagues (2021) elegantly developed a nanoparticle containing the exosomal inhibitor GW4869, hyaluronic acid, and Fe3+ as a ferroptosis inducer [86]. In B16F10 tumor-bearing mice, the administration of this nanoplatform reduced tumor-derived PD-L1 exosomes, increased cell death by ferroptosis, and created an anti-tumor immune response. In addition, the parallel administration of anti-PD-L1 displayed a remarkable tumor regression and robust systemic immune response. This study demonstrated how the knowledge about vesicular secretion could be used for the development of novel strategies to improve tumor response and regression.

## 6. Secretory Autophagy and Exosome Biogenesis in Melanoma: Two Sides of the Same Coin?

As discussed above, the acquisition of resistance in melanoma to different therapeutic modalities is, at least in part, dependent on the vesicular trafficking related to the autophagic, lysosomal, and exosome biogenesis and release. In fact, the similarities among these processes prompt us to hypothesize that the secretion of biological molecules can dictate tumor progression through the phenotype reprogramming of tumor and stromal cells in the TME. Based on this, we searched for common secretory autophagy and exosome-related genes in melanoma specimens using a public database to compare their expression levels with normal melanocytes [87]. Interestingly, these results revealed a higher expression of genes known to be involved in the biogenesis of exosomes (Figure 2a) and SA-derived vesicles (Figure 2b), as well as genes reported as commonly involved in these two pathways (Figure 2c) in melanoma tumors in comparison to melanocytes, reinforcing the idea that both processes are involved in malignant transformation and tumor establishment.

ATG7 is among the genes known to regulate both SA and exosome release. The interaction of ATG7 with ATG5 was reported to induce SA-mediated CFTR release through GRASP-induced CFTR trafficking to MVBs [88]. In addition, ATG7 inhibition was shown to induce MVB disintegration independently of fusion with autophagosomes, thus preventing their exosome secretion [89]. Moreover, as mentioned before, ATG3-ATG12 conjugation by ATG7 was also shown to be required for exosome biogenesis in an Alix-dependent manner [26]. In melanoma, a high expression of this gene was correlated with the increased secretion of IL1β, CXCL8, LIF, FAM3C, and DKK3 in vitro, and metastatic melanoma patients with elevated tumor autophagy were shown to present higher levels of these proteins in their plasma [9].

EPG5, a less noted protein, was first identified as an autophagy-related gene during clinical genetic analysis, which reported that *EPG5* mutation causes a multisystem disorder termed Vici syndrome, characterized by abnormalities in the brain, immune system, and reduced melanin production. Subsequent studies showed that EPG5 acts as a tethering factor being recruited to the lysosomes/late endosomes by Rab-GTPase, which promotes membrane fusion by stabilizing the trans-SNARE complex [90].

RAB8A and STX4 were also shown to play important roles in SA and exosome biogenesis. Regarding secretory autophagy, RAB8A was reported to drive the secretion of IL-1β through interaction with GRASP, while STX4 was shown to induce the fusion of the secretory autophagosome with the plasma membrane through binding with the SNARE complex formed by SEC22, SNAP23, and SNAP29 (Figure 1) [27]. A positive correlation between the expression of these genes and exosome release was shown by different studies. Chen et al. (2017) reported that RAB8A favors the secretion of exosomes containing ANXA2 through the regulation of amphisomes fusion with the plasma membrane [14]. Furthermore, the complex formed by STX4 with SNAP23/29 and VAMP8 has also been associated with exosome release by a mechanism still under investigation [91].

Curiously, melanoma patients present a higher expression of SNAP23 and LC3 after MAPK inhibitors treatment (Figure 2d), although the clinical significance of this finding remains unknown. SNAP23 was reported to promote MVB fusion with the plasma membrane, resulting in CD63-positive exosome release after its phosphorylation on Ser110 [92]; different studies have shown a colocalization of both SNAP23 and LC3 with the exosomal marker CD63 within MVB-like structures in cancer cells [15,93]. Moreover, in silico analysis showed that the expression of SNAP23 and LC3 is correlated with exosome markers (ALIX, CD63, and TSG101) in skin cutaneous melanoma (SKCM) (Figure 2e), supporting the intimate connection between SA and exosome biogenesis and the provocative hypothesis that both processes might represent the same secretory pathway in melanoma. Indeed, proteomics studies available on the Vesiclepedia database demonstrate that distinct proteins involved in autophagy were found in melanoma EVs by different groups (Table 1), including the ones mentioned above as regulators of secretory autophagy (ATG7, GRASP, RAB8A, SEC22, STX3/4, and SNAP23/29) (Figure 1). These data prove that even though SA-derived vesicles are classified as a different mechanism of secretion, they are probably isolated and analyzed along with other subtypes of EVs in melanoma studies. Nevertheless, further effort is required to better understand the interconnection between these pathways or whether they are part of the same secretory route.

## 7. How to Deal with Secretory Autophagy and Exosome Release in Cancer? A Targetable or Not Targetable Pathway: This Is the Question!

Based on the findings about the involvement of autophagy, SA, and exosomes in drug resistance, their pharmacological inhibition has become an attractive alternative to cancer therapy. Pharmacological or genetic inhibition of autophagy has been used in pre-clinical and clinical studies; the pursuit for more efficient and specific autophagy inhibitors is growing. Chloroquine (CQ) and its derivative, hydroxychloroquine (HCQ), are lysosome-autophagosome fusion inhibitors used to impair high levels of autophagy, such as those observed in melanoma cells [94,95]. The use of HCQ/CQ as monotherapy sensitizes melanoma cells and reduces tumor growth [96,97]. Combining these inhibitors with other agents, such as chemotherapeutics compounds, also looks promising based on data from the Rangwala studies. They observed a partial response 3/22 (14%) and stable disease 6/22 (27%) in metastatic melanoma after HCQ+ temozolamide treatment and a better response when they used HCQ + mTOR inhibitor obtaining 14/19 (74%) melanoma patients with stable disease [95,98]. Despite these data, a dozen clinical trials using different strategies are being tested on melanoma, but so far, just two of them have been completed. In this context, it is clear the necessity to pursue more detailed studies with mechanistic results about autophagy modulation, as well as consider the possible induction and modulation of secretory autophagy or exosome release after blocking degradative autophagy. Moreover, the activation of both pathways after treatment may facilitate tumor proliferation, invasion, and resistance.

Regarding exosomes, there are several molecules proficient in reducing exosome secretion. Such compounds act in different stages of the exosome biogenesis and in different proteins—among them are the RAB27A inhibitors and compounds already used clinically, such as tipifarnib, which particularly showed up to be effective in reducing tumor exosome release [99]. The secretion of exosomes by cancer cells is different in number and quality from normal cells [100]. In turn, supporting these data, Parolini et al. reported an increase in exosome release in low microenvironment pH conditions using melanoma cells [101]. Moreover, the exosome traffic in the tumor microenvironment works as a potent path of signaling favoring tumor growth, metastasis, and regulating immune response [102]. Therefore, although still there is no inhibitor used in the clinic, the will for developing strategies to target exosomes is meaningful. Furthermore, we believe that future studies that aim to investigate the tumor response to inhibitors in the context of both exosome/SA-derived vesicle mechanisms are crucial to better understand the regulation and the potential interference between these pathways in cancer.

## 8. Conclusions

In the last decade, the secretion mechanisms and their role in cancer cells have been extensively investigated. Although numerous insights were evidenced, the knowledge about secretory autophagy and the intersection with exosome release is still in construction. Data discussed above show the relevance of these processes in a modulation of melanoma TME. Cargo secretion into the TME by cancer cells may shift cell phenotype sustaining proliferative signaling and regulating immune response.

The advances in the field led to the pursuit for new therapy strategies. So far, clinical studies based on secretion routes are in onset phases and, in fact, a deeper comprehension of these systems is extremely important in order to develop specific drugs and prevent tumor resistance. We considered that for the development of such treatments it is necessary to contemplate the activity of both pathways, thus avoiding possible compensatory mechanisms or negative feedback loops that might lead to tumor recurrence.

## Figures and Tables

**Figure 1 cancers-14-00234-f001:**
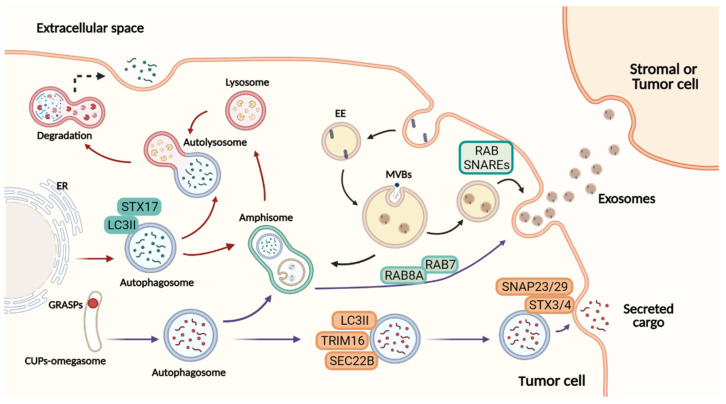
Overview of autophagy-dependent secretory pathways and exosome release. Figure created with BioRender.com.At the ER exit sites, omegasomes contribute to the formation of degradative autophagosomes. LC3II is recruited to the autophagosome membranes, where it mediates membrane expansion. Autophagosomes fuse with lysosomes to form autolysosomes, and their lysosomal hydrolases degrade the cytosolic cargo selected. In a secretory autophagy, the autophagosome is formed by an omegasome-like structure similar to CUPs, near to the ER exit sites. This autophagosome requires TRIM16 for cargo selection, and SEC22B to be addressed to the plasma membrane completing the fusion via complementary SNAREs. Both in the degrading and the secretory pathway, the autophagosome may fuse with the MVBs, generating amphisomes that can either fuse with lysosomes and degrade their content or fuse with the plasma membrane secreting cargo to the extracellular space. Exosome biogenesis initiates from invaginations of the plasma membrane to form early endosomes followed by maturation and inward membrane invaginations to generate ILVs. Further, the MVBs (later endosomes) will fuse with the plasma membrane to extracellularly release the cytosolic components of ILVs as exosomes. The cargo secreted to the extracellular milieu contributes with the transference of several molecules upon the uptake by recipient cells. In this manner, secretory autophagy and exosomes orchestrate multiple systemic processes. Red arrows: degradative autophagy. Purple arrows: secretory autophagy. Black arrows: exosome biogenesis. ER: endoplasmic reticulum. EE: early endosome. MBVs: multivesicular bodies. ILVs: intraluminal vesicles.

**Figure 2 cancers-14-00234-f002:**
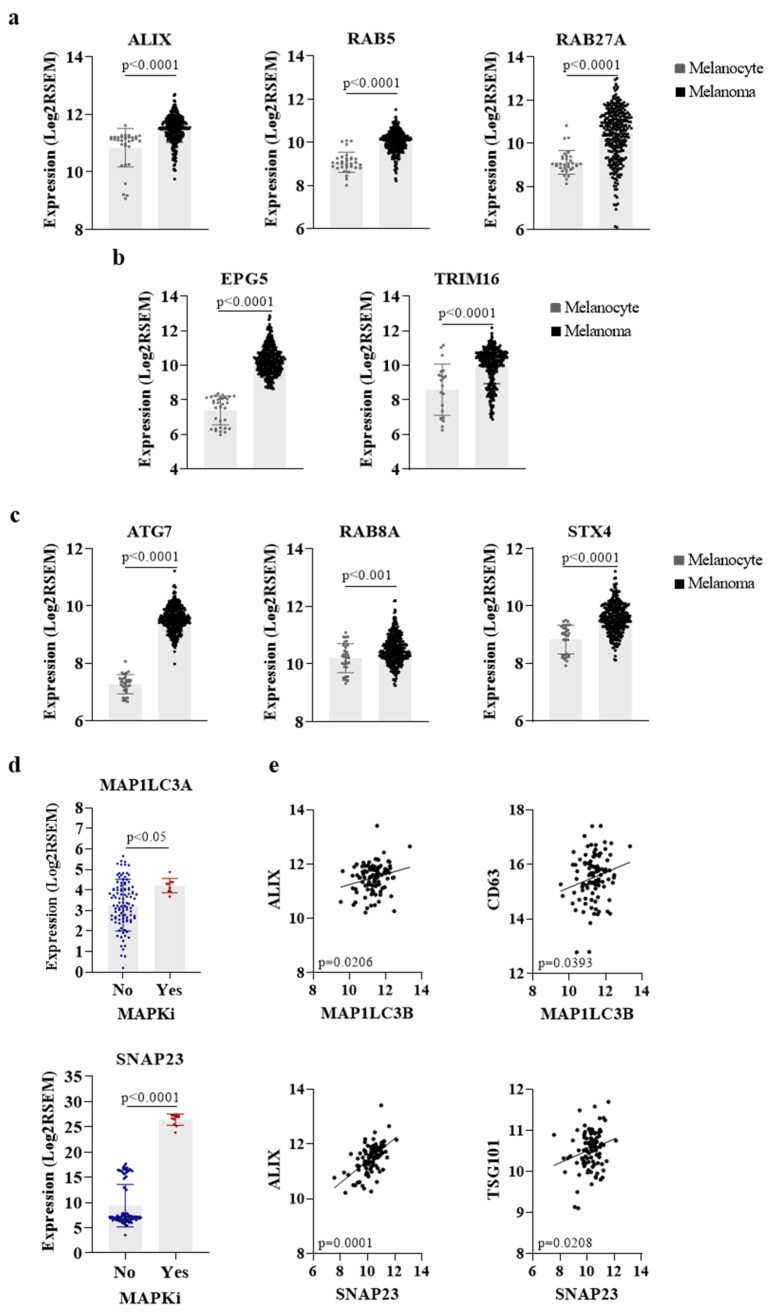
Melanoma presents a higher expression of genes involved in both exosomal and secretory autophagy pathways. (**a**) Expression of genes involved in exosome biogenesis, (**b**) secretory autophagy, (**c**) or both pathways in melanomas compared to non-transformed melanocytes. Gene expression data from melanocytes and melanoma tissues were downloaded from GENT2 platform (GSE30240) and cBioPortal (TCGA/SKCM/20160128), respectively. (**d**) Expression of autophagy-related genes after MAPK inhibitors treatment in SKCM. Data downloaded from cBioPortal (DFCI Nature Medicine/2019). (**e**) Correlation between the expression of LC3 and SNAP23 with exosome markers in SKCM. Data from TCGA/SKCM/20160128. Figures created in Gradphad Prism 9.

**Table 1 cancers-14-00234-t001:** Autophagy-related proteins are found in melanoma EVs.

Protein	Times Identified in Melanoma EVs	Vesiclepedia Experiment IDs
LAMP2	11	(71, 72, 274, 275, 276, 617, 618, 621, 622, 623, 624)
HSPA8	7	(12, 24, 453, 620, 621, 623, 986)
MTOR	6	(617, 618, 619, 620, 624, 625)
PSMD4	6	(617, 620, 621, 622, 624, 625)
ATG7	5	(617, 618, 619, 622, 624)
HSP90AA1	5	(453, 621, 622, 623, 986)
LAMP1	5	(74, 621, 622, 623, 624)
GABARAP	5	(617, 618, 619, 620, 621)
RAB7A	5	(453, 621, 622, 625, 986)
SQSTM1	5	(453, 617, 618, 619, 625)
SEC22A	5	(617, 619, 620, 621, 622)
SNAP23	5	(617, 619, 620, 622, 624)
STX4	5	(617, 618,620, 624, 625)
GARASP	4	(619, 620, 621, 622)
ACBD3	4	(617, 618, 620, 622)
ACBD5	4	(618, 619, 621, 623)
BCL2	4	(619, 621, 622, 624)
PHB2	4	(621, 623, 624, 986)
SNX18	4	(620, 621, 623, 624)
TAX1BP1	4	(453, 622, 623, 625)
TGM2	4	(617, 619, 623, 624)
TRIM16	4	(617, 619, 621, 624)
VCP	4	(453, 622, 625, 986)
SNAP29	4	(617, 619, 620, 625)
RAB8A	4	(617, 619, 621, 623)
EI24	3	(617, 620,624)
EPG5	3	(617, 618, 624)
LGALS3	3	(620, 625, 986)
OPTN	3	(621, 623, 624)
PIK3R4	3	(617, 624, 625)
RAB7B	3	(621, 622, 624)
SNX4	3	(627, 621, 625)
TBK1	3	(617, 619, 622)
TOLLIP	3	(617, 625, 986)
UVRAG	3	(623, 624, 625)
SEC22B	3	(618, 623, 624).
ATG3	2	(618, 620)
BNIP3	2	(619, 621)
RAB11A	2	(621, 622)
SNX3	2	(617, 623)
TFEB	2	(621, 623)
WDFY3	2	(621, 623)
ATG9A	1	(617)
BCL2L13	1	(621)
FUNDC1	1	(625)
GFAP	1	(617)
LGALS8	1	(618)
PEX14	1	(621)
STX3	1	(624)

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
