# Peer review of "Secretory Autophagy Forges a Therapy Resistant Microenvironment in Melanoma"

_cancers, 2022, doi:10.3390/cancers14010234_

Round 1

Reviewer 1 Report

There are several comments on the review:
1. It is recommended to present in more detail the significance of the elements of the tumor microenvironment, especially the role of the vasculature, extracellular matrix, infiltrating lymphocytes and others.
2. Describe in more detail the meaning of Secretory Autophagy for BRAF mutant melanomas.

Author Response

We thank the reviewer for an in-deep analysis of our revision and all the suggestions to improve it. As recommended, we included a new section to better explore the role of autophagy in the tumor microenvironment (section 4. Melanoma progression, tumor microenvironment and autophagy) and better described  the involvment of Secretory Autophagy for BRAF mutant melanomas. These modifications can be found in the new version as it follows below: 

1) section 4. Melanoma progression, tumor microenvironment and autophagy:  "The environment around the tumor is a complex network of stromal cells, including endothelial cells, fibroblasts, and immune cells as well as the non- cellular components such as the extracellular matrix and secreted factors, all important to stimulate tumor heterogeneity. Each element of the TME has essential roles to control several stages of tumorigenesis like tumor growth, invasion, metastasis, and neovascularization. Among them, cancer cells and tumor-associated stromal cells are crucial to alter the equilibrium of extracellular matrix remodeling events, providing a dynamic niche capable of sustaining malignant transformation (Winkler et al,2020). In turn, the extracellular matrix impacts in proliferation, mobility, and tumor vascularization. In the latest, changes in  tissue tumor architecture, caused by overgrowth of cancer cells, aid to the establishment of tumor hypoxia state which ultimately induce a switch to an angiogenic phenotype. Such events lead to the sprouting of new blood vessels vital to deliver sufficient oxygen and nutrients to tumor survival (Siemann et al, 2015, Mae et al, 2016). Above all, the autophagy secretion process has emerged as a regulator of this phenomenon. For instance, the secretion of different cytokines like VEGF-A and IL-8 by melanoma cells trigger pro-angiogenic signaling in surrounding endothelial cells prompting to angiogenesis switch, which is correlated with the transition of melanocytic lesions to vertical growth phase; a stage with higher capacity to metastasize (Thorburn et al, 2009, Dupont et al, 2011, Damsky et al, 2010). The interplay between TME components and secretory autophagy are also crucially involved in local invasion and metastasis, important features in high resistant and metastatic tumors such as malignant melanoma. In this context, it was demonstrated that MMP-9 and MMP-2 in melanoma patients are key players in extracellular matrix degradation and drivers of cancer cell spreading and metastasis (Pandita et al, 2021, Napoli et al, 2020). These peptidases are regulated by different molecules such as the angiogenic factor IL-8 in  an autophagy-dependent secretion manner (Luca et al, 1997), reinforcing the notion that SA establishes an effective communication in the TME. Moreover, another way that autophagy contributes to TME reprogramming is through the regulation of immune cells. Based on the immunogenic phenotype of melanomas characterized by the infiltratation of different immune cells, the activity of the autophagy pathway in this scenario is meaningful. Inhibition of autophagy in melanoma cells, for example, promotes NK (natural killer) cells infiltration and activation against the malignant cells. Interestingly, targeting the autophagy process in MDSC (myeloid-suppressor derived cells) reprograms these cells towards an anti-tumoral phenotype. On the other hand, the secretion of diverse factors into the TME through SA enables the induction of immune evasion and immunosuppression affecting therapy responses and contributing with resistance (reviewed in Liu et al, 2021, Di Leo et al, 2019), as better discussed in the next section. Then, taken together, these studies highlight the oncogenic role of autophagy in the TME in the context of tumor cells as well as stromal cells, and, in fact, we imagine that its blockade specifically in immune cells might improve immunotherapy response and provide clinical benefits to melanoma patients".

2) Concerning secretory autophagy and BRAF melanomas, we have included the following statement: "In fact, most of the studies in autophagy and BRAF mutated melanomas focus on the role of the degradative sub pathway in the acquired resistance which is restricted to the tumor cell. Concerning SA, as described above, there are very few studies about the role of this secretory pathway in BRAF melanomas and we believe that this route may induce resistance not only in tumor cells, but it can also reprogram other stromal cells present in the TME through the secretion of important pro-tumoral molecules. In other words, stromal cells can adopt a behavior triggered by tumor secreted molecules upon targeted therapy, contributing to treatment failure and tumor outgrowth. Besides that, it is also reasonable to consider that important factors released by SA might reach the blood vessels and invoke cellular alterations at longer distances, producing systemic modifications that would help in the dissemination of the disease, for example."

Reviewer 2 Report

In this review, the authors discuss about the cross-talk between the tumor microenvironment (TME) and melanoma cells and how the features of TME play an epistatic role in the regulation of tumor progression. In particular, the authors analyse the role of secretory autophagy or exosomes release, in the communication between the different components of the TME-melanoma circuit. Besides, through the use of public databases, they explore how molecules related to exosome and autophagic pathways affect melanoma progression. The manuscript appears complete and the topic is highly relevant. I found it very interesting and well balanced.

I suggest the authors to further explore the co-regulation between secretory and degradative autophagy (lines 174-180), how this could be cancer specific or a general aspects of cancer evolution. In addition, I found the part described in lines 283-288 about the role of autophagy in chemo-resistance not so clear, the authors should make an effort to better describe the experimental evidence of this aspect. 

Author Response

We thank the reviewer for a careful evaluation of our review.  We appreciate all the observations made by the reviewer and, in the revised version, we better explored the correlation between secretory and degradative autophagy, pinpointing some aspects of cancer evolution under this context. We also rewrote the role of autophagy in chemo-resistance to make it clear as suggested. All the modifications made in the text are described below:

1) Regarding the modifications related to the sentences in lines 174-180, in the revised version, we added the following statement in the end of line 180: "Interestingly, these authors also observed higher levels of autophagy and protein secretion such as CXCL8 and IL1B in metastatic melanomas in comparison to cells derived from primary lesions. CXCL8 has been described as an important regulator of growth, angiogenesis, migration and metastasis in melanomas through its binding to CXCR1 and CXCR2 receptors (Singh & Varney, 1998; Luca et al., 1997; Singh et al., 2010). Consistently, Scheibenbogen and co-authors (1995) showed that higher levels of CXCL8 secreted by melanoma cells correlate with poor prognosis. Concerning IL1B, using a pre-clinical model, Tengesdal et al (2021) demonstrated that IL-1β induces IL-6/STAT3 activation in melanoma, inducing an immunosuppressive environment that favors tumor growth. Then, based on these findings, one might speculate that along melanoma progression, tumor sub clones showing increased levels of SA and EVPs are selected due to their ability in communicating with stromal cells to imprint a permissive microenvironment for tumor growth and metastasis, highlighting the importance of unconventional secretory pathways as an important hallmark in melanoma evolution". 

2) Concerning the sentences in lines 283-288, we rewrote the whole paragraph to make it clearer as it follows: “In fact, concerning the autophagic process, some groups already demonstrated that certain cytotoxic therapies induce autophagy in several tumor types. However, this process can trigger tumor cell death or promote cell survival instead, which seems to be dependent on the cell type and stressor agent. In melanomas, Ma and co-authors [59] verified that a high autophagic flux is associated with poor survival in the metastatic disease. Moreover, analyzing tumor specimens obtained from melanoma patients treated with temozolomide and sorafenib, the same group found that patients harboring tumors with a high autophagic index were less likely to respond to chemotherapy and indeed presented shorter survival, suggesting that targeting autophagy could reverse the chemoresistance in these patients [5]. Focusing on targeted therapy, previous studies revealed that BRAF inhibition upregulates autophagy whereas inhibition of this process overcomes BRAF inhibitor-induced resistance (Goulielmaki et al. 2016; Mulcahy Levy et al., 2017 [5]). However, a recent study showed that fluctuations in autophagy flux occur in response to drug treatment as reported by Verykiou et al (2019). The authors detected that the resistance to MEK inhibitor in melanoma was accompanied by an increase in autophagy in a short term period after trametinib treatment. However, in the long term, this increase was followed by a reduction of autophagy to its basal levels in the surviving cells, indicating that harnessing autophagy can represent a way to bypass induced resistance in metastatic melanoma”.

Reviewer 3 Report

In this review, S. O. Bustos et al. describe the impact of the secretory autophagy in determining a therapy resistant microenvironment in melanoma. This review is an important contribution to understand how this form of autophagy takes part in modulating pro-tumor response on different components of the TME, eventually favoring the development of a resistant microenvironment to therapeutic interventions. Interestingly, the authors discuss on the role of the secretory autophagy and interconnection with the vesicular membrane trafficking and biogenesis using data from public databases.

This reviewer believes that this manuscript is relatively well written and has a well-addressed structure and impact on this topic.

Minor concern:

  • The characters of the bold words in Figure1 are too little.
  • The legend of the Figure1 is erroneously included in the text: see lines 155-157.

Author Response

We thank the reviewer for a careful analysis of our review and a positive reaction towards its impact on the topic. We fixed the issues pointed in Fig 1 and also included this figure correctly in the new version as recommended.